# Effect of Nordic Walking Training on Physical Fitness and Self-Assessment of Health of People with Chronic Non-Specific Lower Back Pain

**DOI:** 10.3390/ijerph20095720

**Published:** 2023-05-04

**Authors:** Mariola Saulicz, Aleksandra Saulicz, Andrzej Myśliwiec, Andrzej Knapik, Jerzy Rottermund, Edward Saulicz

**Affiliations:** 1Institute of Physiotherapy and Health Sciences, The Jerzy Kukuczka Academy of Physical Education, 40-065 Katowice, Poland; 2School of Public Health & Social Work, Queensland University of Technology, Victoria Park Road, Kelvin Grove, Brisbane, QLD 4059, Australia; 3Department of Adapted Physical Activity and Sport, School of Health Sciences in Katowice, Medical University of Silesia in Katowice, ul. Medyków 12, 40-751 Katowice, Poland; 4Health and Social Work, St. Elizabeth University, Namestie 1, maja 1, 811 02 Bratislava, Slovakia

**Keywords:** chronic low back pain, physical activity, fitness, Nordic walking training, quality of health

## Abstract

In order to determine the impact of a four-week cycle of Nordic Walking (NW) training on the physical fitness of people with chronic non-specific lower back pain and the impact of this form of activity on their self-assessment of health quality, the study included 80 men and women aged 29 to 63 years. The subjects were divided into two equal (40-person) groups: experimental and control. In both study groups the degree of disability in daily activities caused by back pain was assessed with the FFb-H-R questionnaire, the physical fitness was evaluated with the modified Fullerton test and the sense of health quality was assessed with the SF-36 questionnaire. The same tests were repeated after four weeks. In the experimental group NW training was applied between the two studies. During four weeks, 10 training units were carried out, and each training session lasted 60 min with a two-day break between each training. The four-week NW training resulted in a statistically significant sense of disability due to back pain (*p* < 0.001), significant improvement of physical fitness expressed by improvement in upper (*p* < 0.001) and lower (*p* < 0.01) body strength, upper and lower body flexibility (*p* < 0.001) and ability to walk a longer distance in a 6-min walk test (*p* < 0.001). The training participants also showed significant improvements in health quality in both physical (*p* < 0.001) and mental (*p* < 0.001) components. The four-week NW training has a positive impact on the physical fitness of men and women with chronic lower back pain. Participation in NW training also contributes to a significant reduction in the sense of disability caused by back pain and improvement in the self-assessment of health quality.

## 1. Introduction

Gait is the basic, natural form of human locomotion. At the same time, it is the most frequently practiced, basic manifestation of physical activity (PA). It can therefore be assumed that from the point of view of human health needs, everyday existence that forces this form of locomotion is also a specific, basic form of training for the body. Nowadays, due to the development of technical civilization, the level of this natural, historically conditioned form of PA and its other types is often insufficient from the point of view of human health needs. Sedentariness, along with its consequences, is a global problem for humanity, posing a threat to public health [1]. This resulted in the need to develop and continuously update appropriate PA guidelines [2]. When it comes to walking, there is a view that the optimal dose for a healthy person is 10,000 steps a day [3,4]. Research reported that people leading a sedentary lifestyle take less than 5000 steps, people who are physically inactive take 5000 to 7499 steps, slightly more active people take 7500 to 9999 steps, while active people take 10,000 to 12,499 steps, and above-average active people take more than 12,499 steps during the day [5].

PA that is beneficial for health should meet a number of criteria. It should be safe, systematically undertaken, and its intensity should result in optimal energy expenditure. It should also positively affect the sphere of human mental functioning. Walking conditioned by meeting the daily needs of human existence most often does not meet their health needs [6]. On the other hand, walking undertaken to improve health can have a beneficial effect on both physical and mental health [7]. However, practicing walking for health reasons can be limited by a number of factors. As with the other forms of PA, the problems might be related to certain personality traits [8], lack of motivation [9], and in the case of people with special needs-functional capabilities, especially the risk of falls [10].

A beneficial form of walking training, both for physical and mental health, is Nordic Walking (NW) [11,12,13]. The main advantages of NW are relative inexpensiveness, accessibility, a sense of security that makes it an inclusive type of activity, and its availability to virtually anyone. There are many reports confirming the preventive and therapeutic effectiveness of NW [13,14,15,16]. It can be particularly effective in eliminating the effects of lifestyle related diseases, especially in middle-aged and elderly people. These conditions also include back pain [17]. Nonspecific low back pain (nsLBP) has a significant contribution to the well-known, global problem of spinal pain epidemiology [18]. According to Walker [19] point prevalence of low back pain (LBP) is 0–30%, the one-year prevalence is 10.3–65% and the life prevalence is 13.8–84%. The average value for point prevalence is 18.3%, for monthly prevalence it is 30.8%, and for lifetime prevalence it is 38.9% [20]. According to Andersson [21] 70–85% of people experience back pain during their lifetime. Low pain intensity associated with slight functional impairment affects 48.9%, high pain intensity associated with slight functional impairment affects 12.3%, and severe functional impairment due to LBP affects 10.7% [22]. 30.6% of people suffering from lower back pain seek help from a doctor [23]. Patients experiencing pain, especially long-term pain, are characterized by a fear of physical activity [24]. In these patients, so-called catastrophic thinking, fear and depression appear [25]. Fear of pain, which may arise when performing certain activities, may change movement habits, resulting in the formation of new ones, which in turn may become the cause of overloads and the source of further pain in other parts of the body [26,27,28,29]. The experience of pain associated with the negative impact of information about the disease results in a pattern of fear avoidance defined Vlaeyen-Linton model [30,31]. Avoiding physical activity in this model is one way to deal with the fear of pain.

The ambiguous etiology, chronic nature of symptoms, psychological effects of recurrent pain and their social implications mean that nowadays LBP is considered based on the biopsychosocial model. In this model, the ability to function, disability or health of a person with LBP are considered not only in terms of functional and structural disorders (e.g., deficit of strength and mobility, degree of degenerative changes) but also in the context of fitness (self-sufficient activities, locomotion), participation and commitment in various life situations (e.g., in the social and professional environment) [32]. In our opinion, PA is an important factor in achieving the goals of this model. It seems that NW can be an effective form of activity, and that resulted in inspiration for the presented study. In the available literature on the subject, no scientific publication evaluating the use of NW training in people with nsLBP has been identified. In addition, there are no reports evaluating the effectiveness of NW in relation to physical fitness and health perception in people with LBP. It was decided to investigate the impact of a 4-week NW training cycle on the physical fitness of people with chronic non-specific low back pain and the impact of this form of physical activity on their self-assessment of quality of health.

## 2. Materials and Methods

### 2.1. Study Design and Subjects

The research covered a group of 80 men and women, inhabitants of the Silesian Voivodeship (Poland). The age of respondents was: 29–65 years. The selection for the study was purposeful, and its criteria were: (a) diagnosed by a specialist doctor (orthopaedist, neurologist, rheumatologist) chronic pain syndrome of the lower spine, persisting for more than half a year, the causes of which could not be clearly determined; (b) another pain episode, lasting at least 12 weeks, of undetermined etiology; (c) no health contraindications to performing moderately intensive physical exercises and intensive walking (d) declared lack of physical activity of a sports-recreational nature. The research was fully voluntary and completely anonymous at the stage of collecting and processing the results.

### 2.2. Randomisation

The subjects, who in the previous therapeutic management were recommended to sustain its effects by maintaining moderate PA, were divided by randomization into two groups: experimental and control. The subjects were randomly divided into groups by a person who neither participated in the study nor conducted the planned walking training. People qualified for the study drew cards with the names “NW” (group undergoing NW training) and “K” (control group). The subjects who qualified for the control group were also offered to participate in NW training, but after the end of the experiment. The surveys and the modified Fullerton Functional Fitness Test (FFFT) were performed in both stages of the experiment by people experienced in performing such tests, who, however, did not know the purpose of the study or the division into groups.

### 2.3. The Course of the Experiment

In the first stage, height and weight were measured and BMI was calculated for all subjects. The level of habitual PA was also assessed using the Baecke questionnaire [33].

At the beginning and at the end of the experiment, the degree of impaired performance in activities of daily living caused by nsLBP was estimated in all subjects. For this purpose, the FFbH-R questionnaire was used to assess the degree of disorders in percentage [34,35]. Overall self-assessment of health was also examined using the SF-36 questionnaire [36]. The physical fitness of the subjects, both at the beginning and at the end of the experiment, was assessed using the FFFT [37,38,39]. This test consists of 5 trials, an assessment of the strength of the upper body (“The Arm Curl”) as well as lower limbs strength (“30-s Chair Stand”), an assessment of upper body flexibility (“Back Scratch”) with lower body flexibility (“Chair Sit-and Reach”) and a 6-min walking test (“The 6-min Walk”) [40]. The walking test was carried out in a closed facility-the march took place along the perimeter of a rectangle measuring 20 m × 4 m. The reason for using this test was the safety of the subjects, possible pain and general fitness level of the subjects.

The experimental group followed a 4-week training program. It consisted of 10 sessions with 2-day breaks between them. Each training session lasted 60 min. The subjects from the control group were asked to lead their usual lifestyle, and in the period of four weeks between the initial and final examinations, they were asked to refrain from strenuous physical activity (including longer walks). The control group was also asked not to undertake new recreational or sports forms of physical activity during this period.

In the experimental group, one person could not participate in all 10 planned training sessions due to random reasons and was therefore not included in the final study. However, in the control group, two people could not report for a follow-up examination due to random reasons, and three people experienced an exacerbation of pain symptoms in the lower spine region. Finally, the results of 39 people from the experimental group and 35 people from the control group were included in the statistical analysis (see Figure 1). The characteristics of the subjects from both groups and intergroup comparisons are presented in Table 1.

Each training session of the experimental group consisted of three phases: a 10-min warm-up phase-preparing the body for exercise, a main phase-lasting 40 min, and a cool down phase-lasting 10 min. The first session was focused on mastering the NW technique. The remaining sessions in the main phase were carried out on flat, wooded and well-paved terrain. Each time during the walking training, a distance of about 5 km was covered. During the walk, a variable pace was used with different stride lengths and intensities of using poles on the basis of interval training. The final stage of the training was a 10-min cooling down phase, in which the walking pace was gradually reduced and interspersed with stretching exercises.

The design of this experiment is part of a larger research project, the implementation of which was planned for many years. The research project on NW application in the population of adults with locomotor dysfunctions as well as physically inactive healthy people was approved by the Bioethics Committee for Scientific Studies at the Jerzy Kukuczka Academy of Physical Education in Katowice (No. 10/2013). All study procedures were performed according to the Helsinki Declaration of Human Rights of 1975, modified in 1983. All participants gave their consent to participate after being informed of the study objectives and procedures.

### 2.4. Statistical Elaboration of the Results

In order to assess the homogeneity of the groups for quantitative traits, the *t*-test for independent samples and the U-Mann-Whitney Test (when the distribution of the traits studied deviated from the normal distribution) and the Chi^2^ test for comparisons of the qualitative traits, were used. The effects of the marching training were assessed using analysis of variance (ANOVA) for repeated measures with the between-subjects factor being group (“NW” vs. control) and within-subjects factor being study (“initial” vs. “final”). When statistical significance for the main effect was achieved, pairwise comparisons were made by using the post-hoc Tukey test. Significance for statistical tests was set a priori at *p* < 0.05.

## 3. Results

A comparison of the two groups (Table 1) showed no statistically significant differences in the basic demographic (gender) and biometric data (age, weight and height, BMI) between the subjects participating in the marching training program and those in the control group. The level of habitual physical activity was also similar in both study groups. On this basis, it can be concluded that both groups were homogeneous in this respect. This allows the influence of gender, body weight and height, as well as overweight and obesity, on the level of physical fitness and self-assessment of the subjects’ quality of health to be disregarded in further analyses.

The mean values of the Functional capacity Score for both groups and the results of the ANOVA test are provided in Table 2. Post hoc analysis showed that both initial and final Functional capacity Score measurements between the two study groups were statistically significantly different. In all cases, higher values (*p* < 0.05 at initial measurements and *p* < 0.001 at final measurements) were recorded in the control group. Before the training, however, these differences were significantly smaller (3.5 points) than after the training (8.4 points). The control group showed no differences between the 1st and 2nd measurements (*p* = 0.639), while a statistically significant decrease in Functional capacity Score (by 6.2 points; *p* < 0.001) after the training was recorded in the NW group.

The mean values of the Fullerton test results for all groups and the results of the ANOVA test are shown in Table 3. Post-hoc analysis showed that both initial and final measurements for ‘The Arm Curl’, ‘30-s Chair Stand’, ‘Back Scratch’, ‘Chair Sit-and Reach’ and ‘The 6-min Walk’ in the two study groups were statistically significantly different. Before the training, however, the intergroup differences were significantly smaller (0.6× for “The Arm Curl”; 2.8× for “30-s Chair Stand”; 1.9 cm for “Back Scratch”, 2.32 cm for “Chair Sit-and Reach” and 2.4 m for “The 6-min Walk” than after the training (3.0× for “The Arm Curl”; 5.3× for ‘30-s Chair Stand’; 8.4 cm for ‘Back Scratch’; 3.51 cm for ‘Chair Sit-and Reach’ and 43.5 m for ‘The 6-min Walk’). The control group showed no difference between the initial and final measurements (for all tests and the walk test *p* > 0.722), while the NW group showed a statistically significant increase in upper and lower body strength tests, an improvement in upper and lower body flexibility and an increase in distance covered by an average of up to 37.5 m during the walk test after the training (*p* < 0.001).

The mean values of the SF-36 questionnaire results and the results of the ANOVA test are presented in Table 4. Post hoc analysis showed significant intergroup differences in both studies for Physical functioning and General Health (*p* < 0.001). In the initial measurements, the experimental group had values higher in Physical functioning by an average of 5 points, while the final measurements showed values higher by an average of 12.6 points. In contrast, for General Health, scores higher by an average of 3.2 points were recorded in the initial measurements in the experimental group, while scores higher by an average of 8.7 points were recorded in the final measurements. In contrast, statistically significant intergroup differences only at the final measurements were registered for Role limitations due to physical health (*p* < 0.01), Pain (*p* < 0.001), Role limitations due to emotional problems (*p* < 0.001), Energy/fatigue (*p* < 0.001), Emotional well-being (*p* < 0.001). Also, for both health components: Physical health component and Mental health component statistically significant intergroup differences were only recorded during the final measurements (*p* < 0.001). Significant differences between the initial and final measurements were registered in the experimental group for all assessed components of the SF-36 questionnaire and the two health components calculated from them (*p* < 0.001). In contrast, in the control group, a post hoc test showed no statistically significant differences between the mean scores of the SF-36 questionnaire completed during the initial and during final measurements (*p* > 0.354).

## 4. Discussion

Regular exercise and intensive walking performed during 10 h of Nordic Walking classes resulted in improved flexibility, both in the upper and lower body. In the final measurements, the participants performed a higher number of repetitions involving bending a weight-bearing forearm and standing up and sitting down on a chair during a 30-s trial, indicating improved upper and lower body strength capabilities.

Parallel to the positive changes in the physical fitness of the women and men taking part in the marching training, a positive effect was registered on the perception of the degree of their disability caused by chronic lumbar pain, expressed by a reduction in their Functional capacity Score of 28.97%. Improved physical fitness, and group interaction during the activities in a relaxing environment (among the greenery in the park) also contributed to an increase in self-assessment of health quality. Indeed, higher values of all components of both the physical component of health and the mental component of health were recorded in the final measurements. The NW training had a slightly greater positive effect on mental well-being, as the Mental health component was rated 22.89% better by the participants in the final survey compared to the initial survey. The improvement in the assessment of the physical health component was slightly lower at 12.94%.

The prolongation of human life, unfavorable demographic trends and the related aging of societies result in the pro-health behavior of middle-aged and elderly women and men is becoming a serious socio-economic problem in developed countries. These problems are aggravated by technological progress and related lifestyle changes with the dominance of passive forms of spending free time. Changing behaviors and habits to the ones that are physically active is an integral part of preventive health programs. Research indicates that health-promoting behaviors targeting physical activity in middle-aged people have a positive impact on their health, while at the same time contributing to the consolidation of positive exercise habits that can counteract the occurrence of modern age diseases or slow down their development [41,42,43,44,45,46,47,48]. From this perspective, physical activity of the adult population of Poles is a cause for concern. Studies conducted in several European countries (Finland, Spain, Germany, Poland, and Russia) have shown that adults in Poland are characterized by the least physical activity [49]. This is a very disturbing fact because, together with diet, physical fitness is one of the basic factors related to the process of so-called successful aging [50]. Physical fitness is a factor that can be modified, which is important in the formulation of pro-health prophylactic programs. The basic factor shaping physical fitness is physical activity. In order to optimize the pro-health effects of physical activity, it is important what kind of measures and what forms of their implementation will be used in practice, especially when preventive measures are applied to physically inactive people who have already experienced the negative effects of hypokinesia. It is important that such an activity should be enjoyable, should bring enjoyment and thus promote motivation for regular exercise. Intense walking with poles seems to be an appropriate form of such an activity. Hayden et al. [51], in a meta-analysis involving the evaluation of the effectiveness of different types of exercise on pain levels and functional limitations caused by lumbar pain, showed that Pilates exercises, McKenzie exercises and functional recovery exercises were the most effective. Nordic Walking based on the natural movements of the human body somehow corresponds perfectly to the ‘functional restoration exercise’.

Some misunderstanding exists concerning the appropriate use of poles when practicing NW. It is believed that, like a cane or elbow crutches, they are used to relieve the lower limbs. Therefore, NW should be recommended especially for people with reduced efficiency of the lower limbs. However, studies have shown that the use of poles while practicing NW does not significantly reduce the load on the knee joints [52]. Poles have not so much a supporting task, but rather their role is to involve the upper part of the body (especially engage the upper limbs and the upper part of the torso) in the walking mechanism. Of course, using poles, especially by less physically fit people, increases the dynamic balance and thus improves the stability of the body while walking. It, therefore, provides some protection against falling, which is of considerable importance for those who have a fear of walking due to past injuries after a fall. An interesting effect of the use of poles during walking was observed in the research by Figard-Fabre et al. [53]. Twelve sessions of walking with poles over four weeks period in 11 obese women resulted in an increase in physiological responses to physical effort at a given speed in these women, but at the same time, their rating of perceived physical exertion was reduced. NW is considered a safe form of physical activity, therefore it is recommended for slightly elderly or sick people. A long-term observation of a total of 29.160 h of NW activities of 101 healthy women and 36 healthy men showed 0.926 injuries for every 1000 h of walking [54]. In comparison, basketball and squash recorded an average of 14 injuries for every 1000 h of their practice. In fact, the only injury specific to NW can be considered a thumb injury. The most common injury was an injury to the ulnar collateral ligament of the thumb, which occurred with a frequency of 0.206/1000 h of walking. Overall, the upper limbs (0.549/1000 h) were more frequently injured than the lower limbs (0.344/1000 h). Muscle injuries mostly affected the m. gastrocnemius (0.137/1000 h). In contrast, there were no injuries to the m. quadriceps femoris and hip muscles. However, more serious injuries such as shoulder dislocation (0.069/1000 h), fractures of the distal part of the radial bone and ankle sprains (0.034/1000 h) are extremely rare. Despite the intense walking characteristic of NW, no injuries to the knee and hip joints were found [54]. Compared to other forms of activity, these data indicate a negligible traumatogenicity of NW. Significantly, Knobloch and Vogt [54] indicate in their study that all people who experienced such an injury returned to their previous physical activity within four weeks at the latest. As a result of the above data, NW, as hardly any other form of physical activity, is suitable for middle-aged and elderly people who were previously not very active, or who were not at all active, and who are particularly susceptible to injury if they undertake such an activity.

The obtained results indicate that the NW training cycle that was carried out over four weeks had a significant impact on the level of physical fitness of the examined group of adults with chronic non-specific lower back pain. Not surprisingly, a significant increase occurred in a distance covered in the 6-min walk test (on average by 37.5 m), since the training sessions consisted mainly of intensive walking, during which a distance of about 5 km was covered on a regular basis. The increase in the strength of the lower limbs is also a consequence of regular walking training. Since intensive arm work is an important element of walking with poles, the increase in upper body strength expressed by a better upper limb strength test in the final measurements should be considered a direct effect of the marching training. The significant improvement in flexibility, on the other hand, is probably related to the form of performance of the individual marching workouts, in which during each meeting the exercises performed in the initial part (shoulder raises, trunk bends and twists, knee lifts) included elements of stretching the longitudinal and oblique musculofascial bands of the upper and lower body. After all, the final part of the class was based on a leisurely walk interspersed with stretching exercises. The positive effects of the form of physical activity used in this study based on intensive walking with poles are not isolated. Positive effects of physical activity carried out in this way have already been demonstrated previously. Rodgers et al. [55] showed that a brisk 30-min march by women with poles increased oxygen consumption by an average of 11%, increased heart rate by 8%, and resulted in an 18% higher energy expenditure compared to brisk walking. In contrast, a study by Porcari et al. [56] of a 20-min march with poles at a submaximal pace, compared to an equivalent walk without poles involving 32 women and men, showed a 23% greater O2 consumption, an 18% increase in heart rate contractions and a 22% increase in energy expenditure. Hagner et al. [57] applied 12-week Nordic Walking training to 168 women divided into three observation groups by menopause (pre, during and post-menopause). Triglyceride levels and BMI were analyzed. Overall triglyceride levels decreased, LDL levels decreased and HDL levels increased, clearly indicating the highly positive impact of this form of physical activity in the context of cardiovascular disease prevention. BMI also decreased significantly. Interestingly, there were no intergroup differences, which means that an equally positive effect was noted among women who were still menstruating, in the middle of menopause or after menopause. Interesting health effects in obese women were recorded by Figard-Fabre et al. [58]. A 12-week NW training program resulted in a significant reduction in body weight and blood pressure in 23 obese women. Studies of the remote effects of walking with poles show that, compared to walking without poles, the greater energy expenditure associated with activating upper body muscles enhances the beneficial effects on the cardiovascular and respiratory systems [59]. In the context of these results, Nordic Walking training is worthy of recommendation for those people to whom physical activity is recommended as a form of weight reduction, improvement of overall physical fitness and prevention of cardiovascular disease. In light of these results, pole walking appears to be an ideal alternative for maintaining physical fitness regained in the process of clinical rehabilitation for people with cardiovascular diseases who additionally suffer from chronic non-specific lower back pain.

According to the modern definition of health, based on a holistic foundation, the focus is on health, not disease. Health is nowadays perceived in three aspects: physical, mental and social. In the holistic understanding of health, the subjectivity of the individual in the pursuit of maintaining it is emphasized [60]. In this view, subjective evaluation of the effectiveness of health-promoting activities becomes important. In our study, along with the objective improvement in physical fitness of the people participating in the marching training program, their subjective assessment of health-oriented quality of life (assessed by the SF-36 questionnaire) significantly improved. The 4-week NW training program resulted in a significant improvement in self-assessment of the physical health component and the mental health component. It is interesting that a greater positive effect was recorded in relation to mental health. The specificity of walking training and, as a result, the improvement of physical fitness of people participating in the experiment would rather suggest a greater impact on the self-assessment of physical health. It is possible that the reasons for this distribution of results could be attributed to the reciprocal feedback between physical fitness and emotional well-being related to self-perception. After all, the study involved people with chronic lower back pain and therefore physically inactive. Decreased physical fitness can lead to reduced needs for social participation, which in turn can translate into the emotional sphere. In other words, a vicious cycle mechanism is at work here, in which a physically inactive lifestyle and the associated reduced physical fitness entails withdrawal from those life situations in which reduced physical fitness may expose one to humiliation in the eyes of those around one. In this view, the perceived improvement in physical fitness may have translated into emotional well-being related to the perception of oneself in relation to others. Hence, perhaps improvements in physical fitness had such a strong impact on the Mental health component. At this stage of the research, however, this is only speculation. If this was the case, NW training would be a good way of breaking down the mental barriers associated with undertaking physical activity for people with chronic pain syndromes. The assessment of the quality of life using the SF-36 questionnaire has already been performed in other studies evaluating the impact of NW training, but it did not involve people with chronic lower back pain. The positive effect of NW training on improving health-oriented self-assessment of quality of life has been reported in patients with peripheral vascular disease [61], chronic obstructive respiratory disease [62], chronic neck pain [63] and in peri-menopausal women [64].

A limitation of this study is the wide age range of the subjects studied, which included both young and advanced middle-aged people. Another limitation is the relatively small study population and the lack of assessment of distant effects. Another limitation was the method of selecting the research material, in which people with chronic LBP who expressed a willingness to participate in NW physical activities were qualified for the study. It is possible that this method of selection resulted in the fact that the research involved mainly people well motivated to return to normal life and probably with a lower level of fear in general and kinesiophobia in particular. Future research should be conducted on a wider population of people with chronic LBP, including the elderly, with the monitoring of the distant effects of NW training. In this evaluation, it is also worthwhile to assess the preventive potential of this form of physical activity to protect against further incidents of acute pain. The assessment of the effects of NW on physical fitness and health perception should be carried out primarily in people with chronic LBP characterized by a high level of fear of movement. In future research, it is also worth comparing NW with other simple forms of PA based on locomotion, such as a longer walk, brisk walking, and even jogging. However, the results of this study indicate that NW can already be recommended to people suffering from chronic non-specific lower back pain as an accessible, low-cost and safe form of exercise.

## 5. Conclusions

A 4-week cycle of NW training helped to significantly improve the fitness level of adults with chronic non-specific lower back pain. The increased physical activity of those taking part in NW activities significantly improved the subjective assessment of health-oriented quality of life. The results obtained in this study allow us to conclude that pole walking according to the NW concept should be promoted as a simple, inexpensive, safe, and, above all, effective form of physical activity that can counterbalance the adverse effects of hypokinesia in adults suffering from lower back pain.

## Figures and Tables

**Figure 1 ijerph-20-05720-f001:**
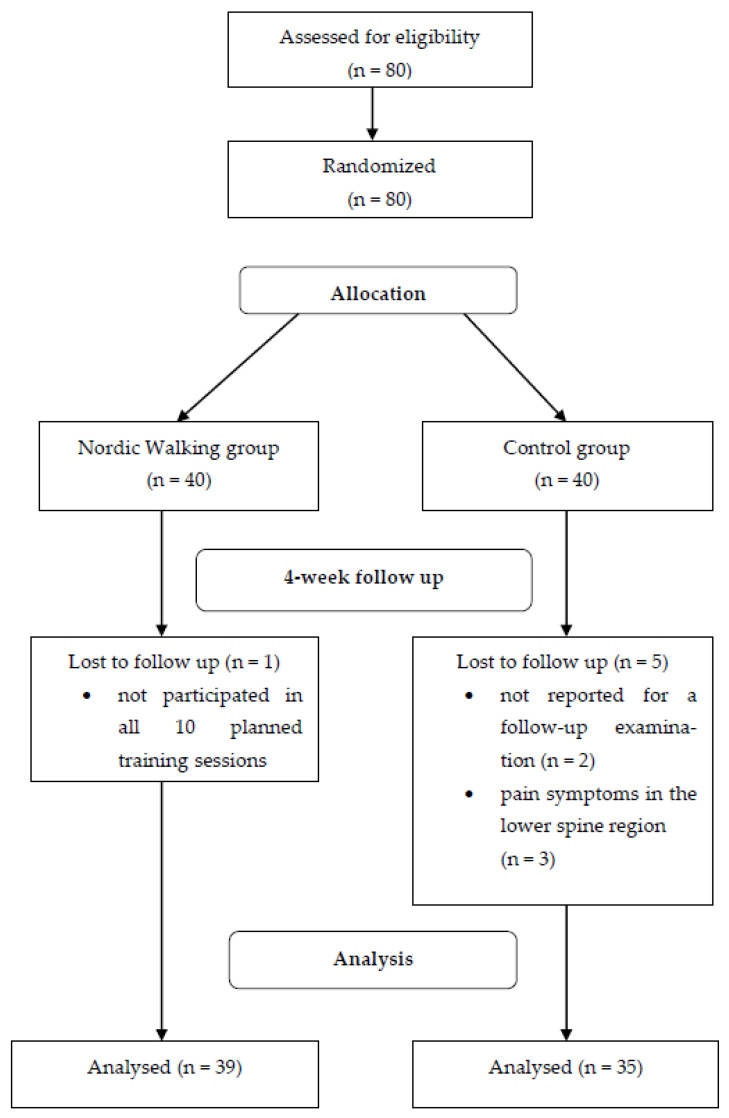
Flow diagram of the study phases.

**Table 1 ijerph-20-05720-t001:** Demographic data of the participants in the Nordic Walking Group and Control Group.

	Nordic Walking Group (*n* = 39)	Control Group (*n* = 35)	*p*
Women (%) Men (%)	31 (79.49) 8 (20.51)	23 (65.71) 12 (34.29)	0.183 ^1^
Age (SD; min-max) [years]	52.5 ± 6.5	50.6 ± 5.1	0.174 ^2^
Weight (SD; min-max) [kg]	71.7 ± 12.6	73.8 ± 12.0	0.466 ^2^
Height (SD; min-max) [cm]	163.6 ± 5.1	163.1 ± 8.3	0.905 ^3^
BMI (SD; min-max) [cm/kg^2^]	26.8 ± 4.2	27.8 ± 4.7	0.303 ^2^
Index of habitual physical activity [pkt]	7.0 ± 1.7	7.2 ± 1.0	0.379 ^3^

^1^ Chi^2^ test. ^2^ *t* test for independent samples. ^3^ U-Mann-Whitney test.

**Table 2 ijerph-20-05720-t002:** Functional capacity score.

Dependent Variables	Nordic Walking Group		Control Group		ANOVA *p* Value		
	Initial Measurement	Final Measurement	Initial Measurement	Final Measurement	Main Effect		Interaction
					Group	Measurement	
Functional capacity score	21.4 ± 14.9	15.2 ± 18.7	24.9 ± 17.8	23.6 ± 16.7	0.127	***	**

* *p* < 0.05; ** *p* < 0.01; *** *p* < 0.001.

**Table 3 ijerph-20-05720-t003:** The results of Fullerton Fitness Test.

Dependent Variables	Nordic Walking Group		Control Group		ANOVA *p* Value		
	Initial Measurement	Final Measurement	Initial Measurement	Final Measurement	Main Effect		Interaction
					Group	Measurement	
Upper limb strength	15.8 ± 3.4	18.0 ± 3.9	15.2 ± 3.8	15.0 ± 4.0	*	***	***
Lower limb strength	16.6 ± 3.9	18.5 ± 3.4	13.8 ± 3.9	13.2 ± 3.8	***	**	***
Upper body flexibility	−7.4 ± 7.6	−0.9 ± 7.7	−9.3 ± 9.1	−9.3 ± 9.0	**	***	**
Lower body flexibility	0.82 ± 5.6	1.81 ± 6.2	−1.5 ± 5.3	−1.7 ± 5.8	*	***	***
The 6-min walk	543.1 ± 80.2	580.6 ± 73.5	540.7 ± 95.1	537.1 ± 95.7	0.273	***	***

* *p* < 0.05; ** *p* < 0.01; *** *p* < 0.001.

**Table 4 ijerph-20-05720-t004:** Results of the physical and mental health component (SF-36).

Dependent Variables	Nordic Walking Group		Control Group		ANOVA *p* Value		
	Initial Measurement	Final Measurement	Initial Measurement	Final Measurement	Main Effect		
					Group	Meaurement	Interaction
Physical functioning	77.3 ± 11.5	85.3 ± 11.6	72.3 ± 19.6	72.7 ± 19.6	*	***	***
Role limitations due to physical health	68.6 ± 39.2	79.6 ± 35.0	60.0 ± 42.9	66.4 ± 40.2	0.211	**	0.460
Pain	56.9 ± 20.1	64.7 ± 20.5	62.2 ± 22.1	63.5 ± 22.2	0.672	***	*
General health	47.5 ± 18.1	53.3 ± 18.6	44.3 ± 14.7	44.6 ± 15.4	0.116	**	*
Physical health component	62.6 ± 19.5	70.7 ± 19.4	59.7 ± 20.6	61.8 ± 20.0	0.193	***	*
Role limitations due to emotional problems	58.9 ± 48.6	82.1 ± 38.9	68.6 ± 45.0	69.5 ± 42.3	0.877	**	**
Energy/ fatigue	45.8 ± 22.0	59.1 ± 22.7	50.9 ± 14.1	51.7 ± 14.2	0.780	***	***
Emotional well-being	52.4 ± 24.8	65.9 ± 20.3	60.1 ± 14.8	60.8 ± 14.5	0.764	***	***
Social functioning	69.9 ± 22.7	72.0 ± 21.3	66.6 ± 19.7	67.4 ± 18.4	0.381	0.368	0.673
Mental health component	56.8 ± 26.6	69.8 ± 23.8	61.6 ± 17.6	62.1 ± 16.8	0.773	***	***

* *p* < 0.05; ** *p* < 0.01; *** *p* < 0.001.

## Data Availability

The data that support the findings of this study are available from the corresponding author, upon reasonable request.

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
