# Peer review of "Effect of Nordic Walking Training on Physical Fitness and Self-Assessment of Health of People with Chronic Non-Specific Lower Back Pain"

_ijerph, 2023, doi:10.3390/ijerph20095720_

Round 1

Reviewer 1 Report

Thank you for opportunity to review the manuscript entitled "Effect of Nordic Walking training on physical fitness and self-assessment of health of people with chronic non-specific lower back pain" 

This is an interesting and well written  study , with convincing size study group. 

In this study authors have tried to  proved the possible role of Nordic Walking as a preventive form of physical acitivity, which can be used in various area of physiotherapy

However, I have major and minor concerns, which should be take into account before publication. 

Major concerns Novelty of the study: for a few years ago Nordic Walking has been very interested subject of scientific investigation. Till now we have many publication concerning  the meaning of Nordic Walking in preventive area and physiotherapy .  More over, many  biomechanical and physiological variables of  NW  were investigated

I wonder what is novel in this study ? It should be highlighted when the hypothesis we be built in Introduction section

In generally - Introduction,  does not supports the topic and hypothesis this study. The authors have  investgated many factors . I am confused, what is the most important in this study :

- influence NW on  functional fitness and mental health or; 

- effects NW in thearapy of low back pain or  relationship between these

In this paragraph authors should explain,  based on literature,  why can we expect the improvement in functional fitness and mental health parmeters after NW, why can we expect the results NW in tratment lor prevention low back pain  

Nevetheless, the main limitation of this study is lack of active  control group . Therfore the results of this study are predictable and are not suprising 

minor concerns:

line 81: no health contraindications 

to perform moderately intensive physical exercises and intensive walking

what does it mean?

line 88-89;  the process of randomization   should be  described in details.  Also flow chart should be created

99-108: what was seting

line 113-114; how was it controlled?

line 122-130;  for readers and researchers detailed information concerning average distance, speed or pace could be very usefull

Reviewer 2 Report

Dear authors, in the attached document you can find the considerations and suggestions for improving your work.
Best regards.

Round 2

Reviewer 1 Report

"Thank you for the opportunity to review your manuscript, I have no further objections."